# Transmissible antimicrobial resistance in *Escherichia coli* isolated from household drinking water in Ibadan, Nigeria

Akeem G. Rabiu[1], Olutayo I. Falodun[2*], Rotimi A. Dada[3], Ayorinde O. Afolayan[4], Olabisi C. Akinlabi[5], Elizabeth T. Akande[5], Iruka N. Okeke[5]

1 Department of Microbiology, Federal University of Health Sciences, Ila-Orangun, Osun State, Nigeria, 2 Department of Microbiology, University of Ibadan, Ibadan, Oyo State, Nigeria, 3 College of Health Sciences, Bowen University, Iwo, Osun State, Nigeria, 4 Institute for Infection Prevention and Hospital Epidemiology, Medical Centre-University of Freiburg, Freiburg im Breisgau, Germany, 5 Department of Pharmaceutical Microbiology, University of Ibadan, Ibadan, Oyo State, Nigeria

* falod2013@gmail.com

## Abstract

Contaminated household water in peri-urban communities is a reservoir for virulent *Escherichia coli*, but its role in the environmental transmission of antibiotic resistance genes (ARGs) remains poorly understood. This study characterized *E. coli* from household water and additionally aimed to investigate the transmissibility of ARGs from drug-resistant isolates. Twenty-five *E. coli* isolates from thirteen household well water sources were tested for resistance to 14 antibiotics by disc diffusion and whole-genome sequenced using the Illumina platform. The ARGs and plasmid replicon types were respectively predicted using ResFinder and PlasmidFinder. Multidrug-resistant strains carrying plasmid replicons found in unrelated strains were conjugated with nalidixic acid-resistant (NAL$^R$) *E. coli* $C_{600}$ using the solid plate method. Fifteen isolates displayed a multi-drug resistance (MDR) phenotype, with 18 possessing ARGs that confer resistance to trimethoprim-sulfamethoxazole, macrolide, sulphonamide, aminoglycoside, chloramphenicol, β-lactams, and tetracycline. Fifteen of the 25 isolates belonged to sequence type detected more than once, and fourteen of these were multidrug resistant. Through solid plate mating, beta-lactam-resistant *qnrS1-tet-dfrA14*-positive strains bearing IncFI-, IncHI2, and IncHI2A successfully transferred ampicillin resistance to a nalidixic acid-resistant derivative of *E. coli*-$_{C600}$. This research highlights the urgent need to safeguard household water sources against fecal contamination to curb the dissemination of ARGs among bacterial populations in this environment.

## Introduction

The spread of antimicrobial resistance (AMR) threatens the efficacy of the antimicrobials administered in clinical practice. In 2019, 4.95 million individuals were estimated

---

**Data availability statement:** All relevant data are within the manuscript and its Supporting Information files.

**Funding:** This work was supported by the African Research Leader's Award MR/L00464X/1 to INO which was jointly funded by the United Kingdom Medical Research Council (MRC) and the United Kingdom Department for International Development (DFID) under the MRC/DFID Concordat agreement and is also part of the EDCTP2 program supported by the European Union. INO is a Calestous Juma Fellow supported by the Bill and Melinda Gates Foundation (INV-036234).

**Competing interests:** The authors have declared that no competing interests exist.

to have died globally due to drug-resistant bacterial infections [1]. AMR surveillance gives greater attention to human clinical isolates and less is known about transmission within and across other One Health sectors (animals and environment). This biased interest misses the opportunity to understand and interrupt the transmission of resistant organisms and/or resistance genes outside the clinic, especially in low- and middle-income (LMIC) countries [2,3]. Household water used for drinking and domestic purposes, particularly in informal urban settlements, is an under-surveilled niche with the potential to impact human health [2]. Examining the role of contaminated water in the environmental transmission of AMR could assist in estimating the degree to which this niche contributes to AMR in settings where good quality water is difficult to access and AMR data are scarce [1]. Moreover, the risk associated with contaminated drinking water as a medium for spreading resistance genes globally is yet to be investigated in detail [4]. Recent model estimates that improvements in Water Sanitation and Hygiene (WASH) could reduce AMR-related deaths by 247,800 [5]. However, the estimates did not include the possibility of non-pathogens in water serving as a reservoir for antimicrobial resistance genes because of a dearth of evidence.

In Nigeria, waterborne diseases are common due to an inadequate supply of clean water, poor sanitation, and, in some cases, the consumption of contaminated household water [6–8]. Alternate water sources to deficient public municipal water infrastructure [9], such as wells and boreholes, particularly when unprotected, used by householders are prone to faecal contamination. Recent research has shown that these sources harbour virulent *E. coli* and other enteric pathogens [8,10,11]. Still, little is known about the contribution of unprotected household water to the bacterial resistance reservoir and the potential of organisms in it to horizontally transfer antimicrobial resistance genes (ARGs).

In southwestern Nigeria, one of the leading antimicrobial-resistant bacterial species found in clinical settings and recovered from contaminated household water is *Escherichia coli* [8,10,11]. Reports demonstrate that extended-spectrum-lactamase (ESBLs) and certain *dfrA* alleles are highly prevalent in African studies, including Nigeria [12,13]. We have recently reported proximal wells of different households can contain genetically indistinguishable strains such that household water is a vehicle for the clonal expansion of resistant bacteria [10]. We theorize that bacteria in households carrying mobile elements may transmit them to other bacteria, particularly in our setting. Therefore, this study was designed to determine whether *E. coli* isolates recovered from contaminated household water are multidrug-resistant and can disseminate the ARGs they carry. We performed antibiotic susceptibility testing (AST) and whole genome sequencing of *E. coli* isolates, inspected the sequences of the strains for ARGs and plasmid replicons, and conjugated them with a plasmid-free *E. coli* strain to explore the transfer of resistance phenotype and genotype.

## Materials and methods

### Study area

The study area is north-east Ibadan in the south-western Nigeria. The sampling sites, Akinyele and Lagelu LGAs, were selected based on their obvious limited or

non-connectivity to Ibadan piped water. These two regions each have been re-demarcated into three municipal districts due to urban renewal: Akinyele East LCDA (AELCDA), Akinyele South LCDA (ASLCDA), Akinyele LGA (ALGA), Lagelu West LCDA (LWLCDA), Lagelu North LCDA (LNLCDA) and Lagelu LGA (LLGA). In urban part of Ibadan metropolis, low water safety and its microbiological quality [14] could have accounted for the incidence of enteric fever and diarrheal disease [8]. However, very little information is available from the peri-urban areas, especially in Akinyele and Lagelu local government areas (LGAs) which are either not covered by the public water distribution network or where residents do not enjoy regular municipal water supply thus warranting a longitudinal and laboratory-based investigation of the water sources for clearer picture [10].

### Strain isolation and identification

In periurban Ibadan, we had reported longitudinal collection of three samples each from 96 household (well = 66 and borehole = 30) water sources which covered a wet and two dry seasons from January 2019 to March 2020 [10] using Raosoft [15] software as the sample size estimator. Here, 25 *E. coli* isolates obtained from 13 households were retrieved from the previous study notably Akinyele East (n = 17), Akinyele South (n = 5), and Lagelu (n = 3) municipal areas [10]. The water sources were previously assessed for microbiological quality including isolation and characterisation of *E. coli* [10] and strains used in this study were from sites with very high coliform and *E. coli* counts (Table 1). Fig 1 shows where the isolates were recovered in each local government area and Table 1 lists all the strains used in the study [10].

### Distribution of isolates in the sampling regions

Twenty-five *E. coli* were obtained from 13 households in Akinyele South (n = 2), Akinyele East (n = 8), and Lagelu (n = 3) municipalities in the peri-urban Ibadan regions. No isolate was retrieved from Akinyele, Lagelu North, and Lagelu West districts. No isolate was also recovered from a borehole: all 25 isolates were retrieved from well water. A median of two *E. coli* isolates per *E. coli*-positive household well water samples was recorded. We previously indicated that fecal contamination of household well water sources in the examined water sources is exacerbated by rainfall [10].

AEW (1–3, 6–9, 11), AS (3, 5), and LAW (2–4) correspond to well water sources in the respective sampling points in Akinyele East, Akinyele South LCDA and Lagelu LGA. Samples were analysed in triplicates. TCC and TEC were the

**Table 1. List of sample sources of *E. coli* used in this study.**

| Sample codes | Source LGA | TCC | TEC | Reference or source |
|---|---|---|---|---|
| AEW1 | Akinyele East | 1750 | 3250 | [10] |
| AEW2 | Akinyele East | 2750 | 1750 | [10] |
| AEW3 | Akinyele East | 6125 | 1250 | [10] |
| AEW6 | Akinyele East | 5750 | 2250 | [10] |
| AEW7 | Akinyele East | 3675 | 1750 | [10] |
| AEW8 | Akinyele East | 1575 | 1250 | [10] |
| AEW9 | Akinyele East | 2150 | 1500 | [10] |
| AEW11 | Akinyele East | 3750 | 1250 | [10] |
| ASW3 | Akinyele South | 2625 | 4250 | [10] |
| ASW5 | Akinyele South | 1450 | 1250 | [10] |
| LAW2 | Lagelu | 16425 | 3250 | [10] |
| LAW3 | Lagelu | 1525 | 5000 | [10] |
| LAW4 | Lagelu | 3375 | 4250 | [10] |
| Control strain | | *E. coli* | | ATCC 25922 |
| $C_{600}$ NAL$^R$ | | *E. coli* NAL$^R$ | | [16] |

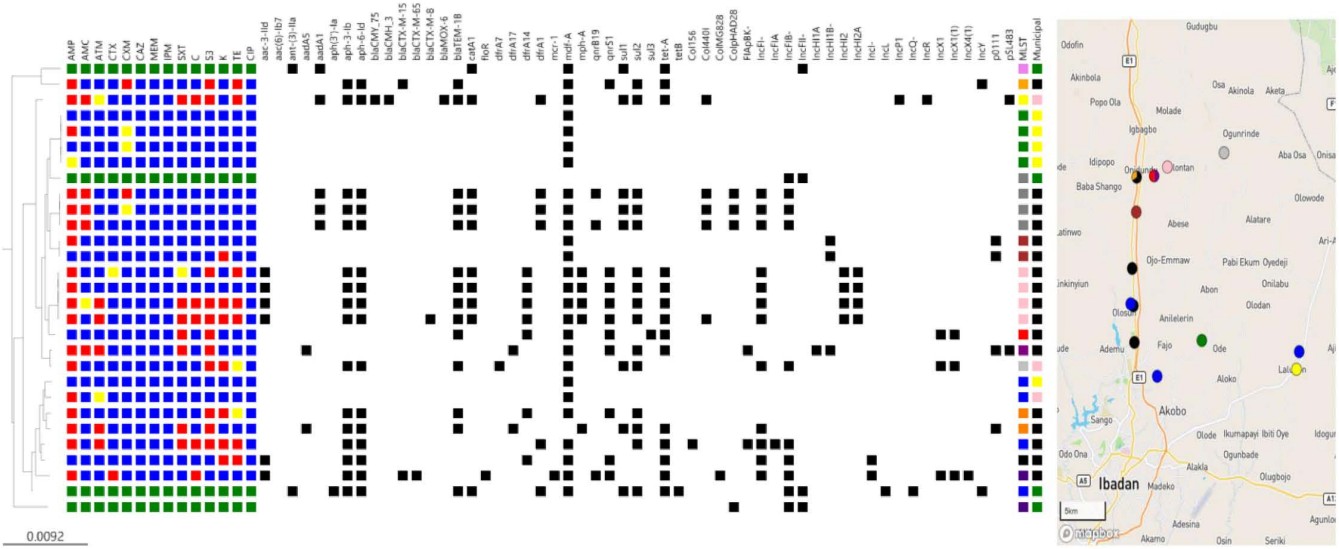

**Fig 1. Whole genome SNP phylogeny and antimicrobial resistance of *scherichia coli* (n =25) isolated from 13 household drinking water in Ibadan, Nigeria.**

sample points' total coliform and total *E. coli* counts, the control strain was *E. coli* ATCC 25922 while the conjugative recipient was nalidixic acid-resistant *E. coli* (*E. coli* NAL$^R$)-$_{C600}$.

## Antibiotic Susceptibility Testing (AST)

The isolates were subjected to AST using the Kirby-Bauer disc diffusion method and results were interpreted following the guidelines of the Clinical and Laboratory Standards Institute [17]. Fourteen antibiotics (Oxoid, United Kingdom) were used: ampicillin, AMP (10 µg), amoxicillin/clavullanate, AMC (10/20 µg), cefuroxime, CFX (30 µg), cefotaxime, CTX (30 µg), ceftazidime, CAZ (30 µg), aztreonam, ATM (30 µg), tetracycline, TE (30 µg), kanamycin, KAN (30 µg), chloramphenicol, C (30 µg), trimethoprim-sulfamethoxazole, SXT (1.25/23.75 µg), sulfonamide, S (250/300 µg), ciprofloxacin, CIP (5 µg), meropenem, MEP (10 µg) and imipenem, IMP (10 µg). *Escherichia coli* American Type Culture Collection (ATCC) 25922 was used as a control.

## Whole genome sequencing

Genomic DNA extraction was performed as outlined previously [10]. Briefly, gDNA was extracted using the Promega Wizard DNA extraction kit and quantified by a dsDNA Broad Range fluorometric quantification assay (Invitrogen). Double-stranded DNA libraries were fragmented, tagged, and sequenced using the HiSeq X10 with 150 bp paired-ends Illumina technology. After genome sequencing, sequence reads were demultiplexed and adapters removed.

**Genome assembly, antibiotic resistance, and plasmid gene identification.** The genomes were assembled *de novo* using Spades v3.9.0 [18] and annotated using Prokka v1.12 [19] with contigs <200 bp and coverage <10-fold excluded from the analyses. The complete sequence of *E. coli*_042 (accession no. FN554766) was used to deduce a whole-genome alignment of the sequence reads, and the SNP positions were determined using the SNP sites [20]. An identity point (90–100%), minimum gene length (60%), and threshold (90.0%) were used to match individual genes for each isolate to the reference database. The prediction of plasmid replicons was done using the PlasmidFinder 2.1 tool on the CGE website [21]. Multilocus sequence typing (MLST) was performed by Achtman's MLST scheme through the website: https://cge.cbs.dtu.dk/services/MLST/. Novel Sequence Types (STs) were assigned numbers by uploading the

raw sequences to Enterobase at https://enterobase.warwick.ac.uk/species/ecoli/upload_reads. The ARGs were predicted using the ResFinder 4.1 tool available on the Center for Genomics Epidemiology (CGE) server (https://cge.food.dtu.dk/services/ResFinder/) [22].

To determine mutation in the quinolone resistance determining region (QRDR) of *qnrS*-carrying isolates; *gyrA*, *gyrB*, *parC*, and *parE* regions encoding fluoroquinolone resistance were called from the raw fastq sequence files using ARIBA [23]. Mutational changes in the called sequences were assessed using PointFinder [24]. The presence of fluoroquinolone resistance-conferring variants was determined by mapping the *gyrA*, *gyrB*, *parC*, and *parE* sequences of the isolates to reference sequence [24]. The PointFinder database was queried to yield (i) point mutation specific to the species, and (ii) a summary of point mutations in the sequences.

### Conjugation on solid media

The *in vitro* bacterial conjugation was performed using the solid plate mating method. Briefly, the donors – the isolates carrying $bla_{TEM}$ and the recipient – nalidixic acid-resistant *E. coli*-$_{C600}$ ($NAL^R$ *E. coli*) were grown overnight in Luria Bertani (LB) broth with appropriate antibiotic selection. The donors were selected with ampicillin or trimethoprim while nalidixic acid was used to select the recipient. Then, a ten-fold dilution of overnight donor and recipient culture was separately grown in LB at 37°C for 1 hr with no antibiotic selection. The donor and recipient cultures, 0.5 mL each, were dispensed into a fresh Eppendorf tube, spun at 6,000 rpm for 5 minutes, and re-suspended in 10 µl LB without antibiotic. The suspension was spotted on LB plates, dried at room temperature for 15 minutes, and incubated at 37°C overnight. The mating culture was later suspended in 1 ml LB, vortexed and the conjugation was stopped on ice; the conjugation was performed in triplicate. A ten-fold dilution of the ended conjugation was prepared in cold phosphate-buffered saline and 10 µl of the reaction culture spread-plated on plates containing the antibiotic combination (i) ampicillin (100 mg/L), trimethoprim (50 mg/L) and nalidixic acid (100 mg/L); (ii) nalidixic acid (100 mg/L) and ampicillin (100 mg/L); (iii) nalidixic acid (100 mg/L) and trimethoprim (50 mg/L). Transconjugants and donors were verified by phenotype on MacConkey and Eosin Methylene Blue agars [12] and re-tested for sensitivity to ampicillin, tetracycline, nalidixic acid, ciprofloxacin, cefotaxime, trimethoprim-sulfamethoxazole, and chloramphenicol. Growth consistent with the parent organisms were taken further. The donor and transconjugant colony-forming units were determined through serial dilutions and results were expressed as transconjugants per donor cell input to calculate the plasmid transfer efficiency [25]. *Escherichia coli* ATCC 25922 was used as the control strain.

### Availability of whole genome sequence data

The raw reads datasets produced in this study were deposited at the European Nucleotide Archive under bio project number PRJEB8667 (https://www.ebi.ac.uk/ena/browser/view/PRJEB8667). Accessions are available in the supplementary file (S1 Table).

### Results

#### Phenotypic and Genotypic resistance of 25 *Escherichia coli* isolated from household water

Antimicrobial susceptibility testing of 25 *E. coli* isolates from household water revealed that all 25 were resistant to erythromycin while majority resisted ampicillin (n = 19/25, 76%). Twelve isolates resisted sulphonamides. Resistance to tetracycline, kanamycin, and trimethoprim was seen in at least five strains. Multidrug resistance (MDR) was observed in 17 isolates, with 15 strains resisting antibiotics that belonged to at least three classes. One isolate was non-susceptible to ampicillin, amoxicillin-clavullanate, sulphonamide, tetracycline, chloramphenicol, and trimethoprim-sulfamethoxazole while two strains, in addition to the latter, resisted kanamycin (Fig 1).

Genotypic analysis unveiled prevalent resistance genes, including aminoglycoside [*aph-6-ld*, *aph-3-lb* (n = 15), *aac-3-lld* (n = 6), *aadA1* (n = 4) *aadA5* (n = 2)], β-lactam [$bla_{TEM-1B}$ (n = 13)], tetracycline [*tetA* (n = 12)], chloramphenicol [*catA1* (n = 8)],

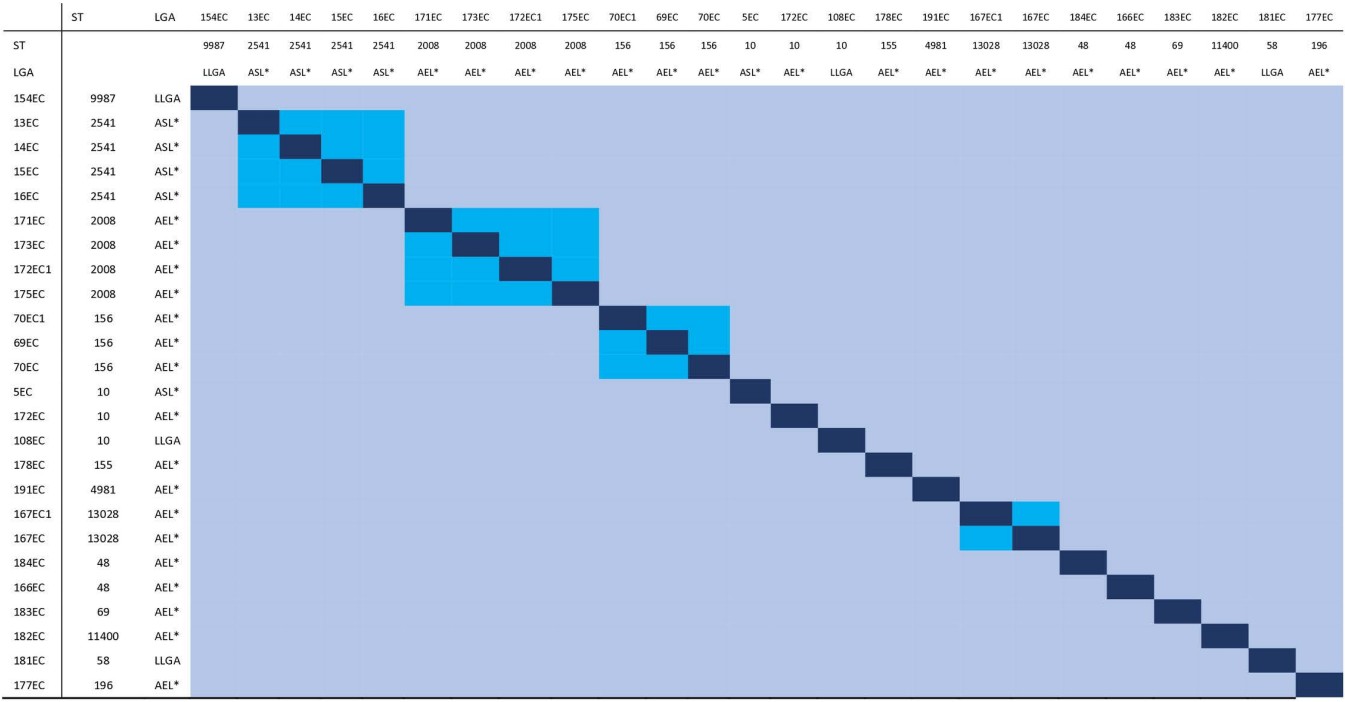

**Fig 2. Matrix showing pairwise Single Nucleotide Difference (SNP) differences between *Escherichia coli* isolates obtained from household well water sources in Ibadan, Nigeria.** Pairs with <10 SNPs are shaded blue whilst > 10 SNPs are shaded light blue. STs and municipal Local Government Areas (LGA) are shaded as for Fig 1. LLGA, AEL* and ASL* connote Lagelu local government area, Akinyele East and Akinyele South LCDA respectively.

trimethoprim *dfrA14* (n = 7), *dfrA1* (n = 5), *dfrA17* (n = 2), and *dfrA7* (n = 1)]. Two variants of *ampC* plasmid-mediated class C β-lactamase genes ($bla_{CMY-75}$ and $bla_{CMH-3}$) and a $bla_{MOX}$ gene were found in a $bla_{TEM-1B}$-positive strain. Extended-spectrum-lactamase (ESBL) genes $bla_{CTX-M-15}$ and $bla_{CTX-M-65}$ co-existed in the genome of one strain, while two additional separate strains carried one of these ESBL genes: $bla_{CTX-M-8}$ and $bla_{CTX-M-15}$. *sul1, sul2,* and *sul3* were respectively borne by six, 14, and one isolate(s). Isolates bearing *sul1* bore *sul2* but not vice versa while the *sul3* was not detected with any other *sul* genes. No isolate carried *aac-6-lb-cr*, which confers ciprofloxacin and aminoglycoside resistance (Fig 1).

Although no carbapenemase genes were detected, nor was resistance to carbapenems, one multidrug-resistant strain carrying mobile colistin resistance gene *mcr-1* was found. Resistance genotypes detected in the isolates are likely responsible for their β-lactams, sulphonamide, and trimethoprim-sulfamethoxazole resistance. There was, however, discordance in aminoglycoside, tetracycline, and chloramphenicol phenotype-genotype. For instance, seven out of 12 isolates carrying *tet*, and five out of nine *catA1*-positives showed resistance (Fig 1). The topoisomerase IV subunit A *parC* genes of the isolates have about 99% percentage identity with the reference sequence [22]. Strains [182EC, 183EC, 184EC; SNP = 15] and [172EC and 173EC; SNP = 23] tested sensitive to quinolones even though they bore the *qnrS1* PMQR gene.

While 18/25 *E. coli* carried at least one plasmid replicon; isolates that showed resistance to more antimicrobials, particularly those harbouring ESBL genes, carried more plasmid replicons. For instance, *E. coli* (191EC) isolate carrying *mcr-1* and other ARGs [*aph-6-Id, aph-3-Ib, aac-3-Iid,* $bla_{CTX-M-15}$, $bla_{CTX-M-65}$, *floR, qnrS1, tetA*] bore six replicons – ColMG828, IncFI-, IncFIB, IncX1, IncX1(1), IncX4(1) and IncI. It was observed that isolates (n = 16/18) carrying plasmids were isolated from north-eastern Ibadan (Fig 1). Three plasmid replicon types (Col-, IncF-, and IncHI-) were predominant and accounted for more than 60% of all the replicons. Five major STs were detected in this study - ST10 (5EC, 108EC, 172EC), ST2541

(13EC, 14EC, 15EC, 16EC), ST2008 (171EC, 172EC1, 173EC, 175EC), ST156 (69EC, 70EC, 70EC1), ST48 (166EC, 184EC) and novel ST13028 (167EC, 167EC1). Other strains individually belonged to STs-58, 69, 155, 196, 4981, 11400, and 9987*. Except for four ST2541 strains, which were near pan-sensitive, and three ST156 multidrug-resistant isolates with remarkably similar resistance gene (*aadA1*, *aph-3-Ib*, *aph-6-Id*, *bla*$_{TEM-1B}$, *catA1*, *dfrA1*, *qnrB19*, *sul1*, *sul2*) and IncF plasmid replicon, closely related strains belonging to the same ST typically had different resistance gene and plasmid replicon profiles. However, the diversity of the strain set was considerable and there were only six such clusters. Certain resistance gene combinations were seen repeatedly. Six of the seven strains with dfrA*14*, which carried a range of plasmid replicons (n = 8) or no detected plasmid replicon (n = 1), also bore *tetA* and *qnrS1*, whilst four of five strains bearing *dfrA1* carried *qnrB19*, *sul1* and *sul2* and carried IncF plasmid replicons. Furthermore, the strains were screened for a few bacteriocin genes common among *E. coli*. These include *cia*, *cba*, *cma*, *cvaC-*, *cib*, *mchF*, *mchB*, *mchC*. No colicin/ bacteriocin was detected more than once per isolate, and 22 of the isolates did not possess any of these genes.

## Spatial overview of isolate lineages resistance genes and plasmids

A total of 15 multilocus sequence types were found among the 25 isolates with six STs recovered more than once. There were four isolates each belonging to STs 2008 (all from AELCDA), and 2541 (all recovered in ASLCDA), three ST156 isolates (from AELCDA). All isolates belonging to the same ST that had identical or near-identical resistance gene profiles were isolated from wells in the same LGA located less than 1 km apart. The three ST156 isolates recovered in AELCDA carried [*aadA1*, *dfrA1*], and two carried *qnr1b*. These strains carried IncF [IncFI- and IncFIB-], which were also found in multidrug-resistant isolates from proximal sites in AELCDA (Fig 1). The four ST2008 isolates carried (*bla*T$_{EM-1B}$, *bla*$_{CTX-M-8}$, *catA1*, *dfrA14*, *mph-A,* *qnrS1*, *sul2*, and *tetA* which were associated with IncF and IncH plasmid replicons. Two non-clonal ST10 isolates carrying IncF and IncH plasmid replicons and similar resistance gene profiles were also isolated from AELCDA, at locations proximal to the multidrug-resistant ST2008 strains. In addition to the IncH replicon-bearing strains, there were clusters of different STs that bore similar plasmid replicon and resistance gene repertoires. Most notable among these were IncX-bearing strains from STs 58, 196, and 4981, which carried among other ARGs (*aph-6-Id*, *aph-3-Ib*, *aac-3-lid*, *tetA*, *qnrS1*, *bla*$_{TEM-1B}$, *dfrA14*) ESBL *bla*$_{CTX-M-15}$, *bla*$_{CTX-M-65}$, *floR* and plasmid-mediated colistin resistance gene, *mcr-1*.

## Incidence of *Escherichia coli* across and within households

The clonality of the strains was investigated through Single Nucleotide Polymorphism (SNP) analysis to understand isolate transmission within and across the 13 households where the organisms were found. There were two ST48 strains and four ST10 isolates but these had different resistance and plasmid replicon profiles and differed by >10,000 SNPS. As shown in Fig 2, in ASLCDA, there was a cluster of four, predominantly sensitive ST2541 isolates from the same household (13EC, 14EC, 15EC, and 16EC), which differed by only 2–4 SNP differences. Three ST2008 (171EC, 173EC, 175EC) recovered from two neighboring AELCDA households carried identical ARGs [*dfrA14, aac-3(lid), aph-3(Ib), aph_6(Id)*, *bla*$_{TEM_1B}$, *catA1*, *mphA*, *qnrS1*, *sul2*, *tetA*] and IncH replicons, and shared zero SNPs. Elsewhere in AELCDA, ST156 strains 69EC, 70EC, and 70EC1, harbouring identical ARGs were isolated from two households in the same neighborhood. These strains had no SNP differences among them and, like the ST2008 isolates were therefore likely derived from a common ancestor.

Red/blue colours respectively depict the presence/absence of the corresponding resistance phenotype, according to CLSI standards. Intermediate phenotypes are coloured yellow and green implies Not Applicable. The ARGs and plasmid replicon markers present are shown in black. Of the six municipal areas, isolates carrying plasmid replicons were retrieved from three municipalities: ASLCDA (yellow), AELCDA (black), and LLGA (pink). The colours white/black respectively illustrate the absence/presence of the corresponding ARGs. The sequence types were

colour-annotated such that ST10 = blue; ST2541 = green; ST9987 = yellow; ST48 = cream; novel ST13028 = brown; ST2008 = pink; ST196 = red; ST155 = purple, novel ST11400 = black; ST69 = orange; ST4981 = indigo; ST156 = peach; ST414 = violet; ST11 = grape and ST9005 = grey. The isolates are represented on the Microreact Map based on MLST colour coding. The majority of the isolates (n = 18/25) carried plasmid replicons whereas most of the plasmid-bearers (n = 16/18) were found in isolates obtained from the north-eastern part of Ibadan, especially the Akinyele East municipal area (AELCDA). The data can be visualised interactively on Microreact at https://microreact.org/project/bsddEhN1iRsb6rdGvmA5Yk-genomic-data-of-e-coli-isolated-from-well-water-sources-in-ibadan-nigeria.

## Conjugation of *Escherichia coli* strains with *E. coli-C600*

To explore the possibility of ARGs horizontal transmission, six isolates multidrug-resistant isolates not showing phenotypic quinolone resistance, were mated with a nalidixic acid-resistant variant of strain *E. coli*-C600. As shown in Table 2 ampicillin resistance, but not other resistances, were transferred *in vitro* from two of the strains. The efficiency of ampicillin resistance gene transfers for 171EC and 183EC respectively were $1.49 \times 10^4$ and $2.2 \times 10^3$ (Table 2). Trimethoprim resistance was not transferred under the experimental conditions.

## Discussion

In this study, *E. coli* strains recovered from household well water were subjected to antimicrobial susceptibility testing and 15 (of 25) of these isolates were multidrug-resistant, carrying genes conferring resistance to three or more clinically significant antimicrobial classes. Resistance to commonly used 'Access' antimicrobials was frequently seen, but resistance to Watch/Reserve antimicrobials (cefotaxime, cefuroxime, ceftazidime, imipenem, meropenem, ciprofloxacin) was less common although plasmid-mediated quinolone resistance was often detected and *mcr-1*, encoding colistin resistance was detected in one strain. This finding and the extent of the MDR *E. coli* (n = 15, 60%) is alarming because the prevalence and types of antimicrobial resistance genes (*dfrA*, *bla*$_{TEM}$, *bla*$_{CTX-M}$ and *qnrS1*) are similar to what is reported in clinical isolates in Nigeria [8,26,27]. It is worrying that these antimicrobial resistance genes are capable of thwarting treatment of infections caused by these strains, especially when first-line antimicrobials are administered.

**Table 2. Result of conjugation experiment and efficiency of plasmid transfer.**

| Donors | Resistance Phenotype | Resistance Genotype | Plasmid replicons | Transconjugants' resistance phenotype | AMP transfer efficiency |
|---|---|---|---|---|---|
| 171EC – ST 2008 | AMP, C, S3, TE | *bla*$_{TEM1B}$, *catA1*, *mphA*, *mphB*, *sul2* * | IncFI-, IncHI2, IncHI2A | ampicillin | $1.49 \times 10^4$ |
| 172EC1 – ST10 | AMP, ATM, SXT, C, S3, K, TE | *bla*$_{CTX-M-8}$, *bla*$_{TEM1B}$, *catA1*, *mphA*, *sul2* * | Col440I, IncFI-, IncHI2, IncHI2A | nil | nil |
| 173EC – ST 2008 | AMP, SXT, S3, TE | *bla*$_{TEM1B}$, *catA1*, *mphA*, *sul2* * | IncFI-, IncHI2, IncHI2A | nil | nil |
| 175EC – ST2008 | AMP | *bla*$_{TEM1B}$, *catA1*, *mphA*, *mphB*, *sul2* * | IncFI-, IncHI2, IncHI2A | nil | nil |
| 177EC – ST196 | AMP, SXT, S3 | *bla*$_{TEM1B,}$ *mphB*, *mef-B*, *sul3* | IncX1, IncX1.1 | nil | nil |
| 183EC** | AMC, K, TE | *bla*$_{TEM1B}$, *bla*$_{CTX-M-15}$, *qnrS1*, *tetA*, *sul2* | IncY | ampicillin | $2.2 \times 10^3$ |
| *E. coli* ATCC 25922 | NA | NA | NA | nil | – |

AMP, ampicillin; C, chloramphenicol; S3, sulphonamide; TE, tetracycline; ATM, aztreonam; SXT, trimethoprim-sulfamethoxazole, CXM, cefuroxime; CTX, cefotaxime; K, kanamycin; NA, Not Assessed; *Isolates carrying *aac-3(IId)*, *aph-3(Ib)*, *aph-6(Id)*, *qnrS1*, *tetA*, and *dfrA14* genes

Antibiotic resistance phenotypes and genotypes for ampicillin, erythromycin, sulphonamide, and trimethoprim-sulfamethoxazole, but not for ciprofloxacin and tetracycline, were concordant. Most multiply-resistant isolates resisted tetracycline, ampicillin, trimethoprim-sulfamethoxazole, amoxicillin/clavulanic acid, and chloramphenicol. These Access antibiotics are cheap and orally active, commonly used at the primary care level in Nigeria [28]. The isolates were not resistant to ciprofloxacin, but PMQR genes [*qnrS*1 (n=6), *qnrB81* (n=1), *qnrB19* (n=3)] were common. These genes are insufficient to confer quinolone resistance, but can, however, contribute to ciprofloxacin resistance when present in combination with other quinolone resistance genes or mechanisms [29]. The absence of QRDR mutations in the strains carrying these genes accounts for why resistance was not seen but transfer of the PMQR genes can cause clinically significant quinolone resistance in a recipient strain that has them [12]. In this study, *qnrB81* and *qnrB19* genes co-occurred with ColpHAD28 and Col440I plasmid replicons, which have been shown by other workers to mediate transfer and confer decreased susceptibility to quinolones [30,31].

Studies by Sumrall *et al*. [12] and Fortini *et al*. [32] in Nigeria indicated that the *qnrS1* allele is associated with *dfrA14* and *tetA* genes flanked by insertion sequences and carried on IncX plasmids. In this study, six isolates were *qnrS1*-positive but none carried an IncX replicon: the gene repertoire co-occurred with IncFI, IncHI2, IncHI2A, and IncY replicons. Inspection of the genome sequences for *dfrA-tet-qnrS1* revealed that *qnrS1* was commonly seen in the genomes of multidrug-resistant strains while *dfrA14* and *tetA* were co-founded with *qnrS1* in four out of the six *qnrS1*-positive MDR isolates. Of the three strains harbouring the *qnrB19* gene: one bore *tetA* while the others carried *dfrA1*. It is noteworthy that none of the strains carrying *qnrB19* simultaneously bore *tet* or *drfA* genes.

Previous studies in Akinyele that examined animal fecal matter reported isolates resistance to β-lactam (penicillin and ampicillin) and β-lactam inhibitor antibiotics (amoxicillin/clavulanic acid) [33]. Clinical human *E. coli* isolates from the same setting have also shown phenotypic resistance to similar (β-lactam – ampicillin) and other antibiotics – trimethoprim/sulfamethoxazole and ciprofloxacin [8,27,34]. Furthermore, the current study reports $bla_{TEM-1B}$, ESBL $bla_{CTX-M-8}$, $bla_{CTX-M-15,}$ and $bla_{CTX-M-65}$ and trimethoprim [*dfrA1*, *dfrA7*, *dfrA14,* and *dfrA17*] all of which have been detected in recent clinical studies [8,27,34] suggesting that these strains likely oscillate between humans and the environmental settings.

A wide range of multilocus sequence types was detected among the isolates but there were four ST2008 and two ST48 strains isolated from the same local government area that were multidrug-resistant and other STs that showed within ST similarity in resistance gene profiles. In the case of two ST2008 and the ST156 isolates, the plasmid replicons were also similar, suggesting that these clusters, comprised of strains from different households may be derived from a point source even though a distance of about 500 metres lay between the wells from which the different isolates were recovered. SNP difference evaluations in Fig 2 show that, ST2008, ST 2541 (n=4), ST 156 (n=3), and ST 13028 (n=2) circulating in AELCDA and ASLDA, where most of the isolates were recovered, were clonal sharing identical ARGs and plasmids (Fig 1). In the case of ST2008 and ST156 clusters, clonal isolates were recovered from water sources of different but neighboring households within the Akinyele sampling region [Akinyele East and Ikereku].

In addition to pointing to the possible clonal transmission of multidrug-resistant *E. coli* strains, the finding of plasmid and resistance gene profiles in resistant clones that were seen in unrelated strains from other STs pointed to possible mobility of key resistance genes, something that has been previously reported for fecally derived isolates in Nigeria [12]. Our study included 11 isolates bearing the resistance gene combination *dfrA-tet-qnrS1*. As stated above, these strains did not bear IncX plasmid replicons as found by Sumrall et al [12] but instead carried IncF and IncH replicons and all carried ESBL genes. When we mated six of these isolates with a nalidixic acid-resistant plasmid-free *E. coli*-$_{C600}$ strain, resistance to trimethoprim, tetracycline, or the quinolones was not transferred but beta-lactam resistance was in two cases. The observed conjugation ESBL frequencies of 1.49 x $10^{-4}$ and 2.2 X $10^3$ were high enough to suggest that horizontal gene transfer could be possible outside the laboratory, and even in the household water ecosystem, at least for ESBLs.

## Conclusion

*Escherichia coli* isolated from different household water sources in Ibadan carry a range of antimicrobial resistance genes, some of which can be transmitted horizontally, constituting a health threat and risk of infection with fecal-oral

transmitted diseases. Improvements in Water, Sanitation, and Hygiene can potentially impact the transmission of antimicrobial resistance and enteric infections. This study unveils that without preventing fecal contamination of drinking water, the spread of fecally derived bacteria can support the horizontal spread of ARGs in household water by clonal expansion of resistant strains and transmission of mobile genetic elements. The small number of isolates examined limited a wider generalisation of the import of the study. Thus, further studies are suggested to determine whether household drinking water can lead to the human acquisition of environmentally disseminated ARGs. This would help to evaluate the extent of the threat to public health posed by consuming drinking water contaminated with bacteria carrying transmissible genes. In the interim, sensitising the householders on the use of ceramic and biosand water filters [35] is recommended to forestall the spread of microbes carrying mobile and transmissible resistance traits.

## Supporting information

**S1 Table.**   Information on the isolates' bioproject, sample, and sequence read archive accession.
(XLSX)

## Acknowledgments

None

## Author contributions

**Conceptualization:** Akeem G. Rabiu, Olutayo I. Falodun, Iruka N. Okeke.

**Data curation:** Akeem G. Rabiu, Rotimi A. Dada, Ayorinde O. Afolayan, Elizabeth T. Akande, Olutayo I. Falodun, Iruka N. Okeke.

**Formal analysis:** Akeem G. Rabiu, Rotimi A. Dada, Ayorinde O. Afolayan, Elizabeth T. Akande, Olutayo I. Falodun, Iruka N. Okeke.

**Funding acquisition:** Iruka N. Okeke.

**Investigation:** Akeem G. Rabiu, Rotimi A. Dada, Ayorinde O. Afolayan, Olabisi C. Akinlabi, Elizabeth T. Akande, Olutayo I. Falodun, Iruka N. Okeke.

**Methodology:** Akeem G. Rabiu, Rotimi A. Dada, Ayorinde O. Afolayan, Olabisi C. Akinlabi, Elizabeth T. Akande, Olutayo I. Falodun, Iruka N. Okeke.

**Project administration:** Olutayo I. Falodun, Iruka N. Okeke.

**Resources:** Akeem G. Rabiu, Rotimi A. Dada, Ayorinde O. Afolayan, Olabisi C. Akinlabi, Elizabeth T. Akande, Olutayo I. Falodun, Iruka N. Okeke.

**Software:** Akeem G. Rabiu, Rotimi A. Dada, Ayorinde O. Afolayan, Olabisi C. Akinlabi, Elizabeth T. Akande, Iruka N. Okeke.

**Supervision:** Olutayo I. Falodun, Iruka N. Okeke.

**Validation:** Akeem G. Rabiu, Rotimi A. Dada, Ayorinde O. Afolayan, Olutayo I. Falodun, Iruka N. Okeke.

**Visualization:** Akeem G. Rabiu, Rotimi A. Dada, Ayorinde O. Afolayan, Iruka N. Okeke.

**Writing – original draft:** Akeem G. Rabiu.

**Writing – review & editing:** Akeem G. Rabiu, Rotimi A. Dada, Ayorinde O. Afolayan, Olabisi C. Akinlabi, Elizabeth T. Akande, Olutayo I. Falodun, Iruka N. Okeke.

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
