## [Decision Letter · Decision Letter 0]

26 Feb 2025

PONE-D-25-04412Transmissible antimicrobial resistance in Escherichia coli isolated from household drinking water in Ibadan, NigeriaPLOS ONE

Dear Dr. Falodun,

Thank you for submitting your manuscript to PLOS ONE. After careful consideration, we feel that it has merit but does not fully meet PLOS ONE’s publication criteria as it currently stands. Therefore, we invite you to submit a revised version of the manuscript that addresses the points raised during the review process.

We look forward to receiving your revised manuscript.

Kind regards,

Mabel Kamweli Aworh, DVM, MPH, PhD. FCVSN

Academic Editor

PLOS ONE

Journal Requirements:

2. Thank you for stating the following financial disclosure: [This work was supported by the African Research Leader’s Award MR/L00464X/1 to INO which was jointly funded by the United Kingdom Medical Research Council (MRC) and the United Kingdom Department for International Development (DFID) under the MRC/DFID Concordat agreement and is also part of the EDCTP2 program supported by the European Union. INO is a Calestous Juma Fellow supported by the Bill and Melinda Gates Foundation (INV-036234)].

3. Thank you for stating the following in the Acknowledgments Section of your manuscript: [This work was supported by the African Research Leader’s Award MR/L00464X/1 to INO which was jointly funded by the United Kingdom Medical Research Council (MRC) and the United Kingdom Department for International Development (DFID) under the MRC/DFID Concordat agreement and is also part of the EDCTP2 program supported by the European Union. INO is a Calestous Juma Fellow supported by the Bill and Melinda Gates Foundation (INV-036234).]

Please remove any funding-related text from the manuscript and let us know how you would like to update your Funding Statement. Currently, your Funding Statement reads as follows: [This work was supported by the African Research Leader’s Award MR/L00464X/1 to INO which was jointly funded by the United Kingdom Medical Research Council (MRC) and the United Kingdom Department for International Development (DFID) under the MRC/DFID Concordat agreement and is also part of the EDCTP2 program supported by the European Union. INO is a Calestous Juma Fellow supported by the Bill and Melinda Gates Foundation (INV-036234)].

Additional Editor Comments:

In addition to addresing the concerns of the reviewers, kindly fix the following issues;

1. In the last paragraph of the Discussion section, please highlight the **key limitations ** of this present study

2. Lines 334–340 provide a summary of the study and future directions. Please merge these with the conclusion, ensuring that the conclusions appear first, followed by the future directions or recommendations.

3. Upon reviewing your reference list, we noticed that **50% of your citations are older than five years** , whereas we recommend that **no more than 20% of references be older sources** unless they are foundational studies. To align with this guideline, please **update your reference list by incorporating more recent literature from the past five years** while ensuring that older references are limited to key foundational studies.

Reviewers' comments:

Reviewer's Responses to Questions

**Comments to the Author**

1. Is the manuscript technically sound, and do the data support the conclusions?

Reviewer #1: Yes

Reviewer #2: Yes

Reviewer #3: Yes

2. Has the statistical analysis been performed appropriately and rigorously? 

Reviewer #1: N/A

Reviewer #2: N/A

Reviewer #3: N/A

3. Have the authors made all data underlying the findings in their manuscript fully available?

Reviewer #1: Yes

Reviewer #2: Yes

Reviewer #3: Yes

4. Is the manuscript presented in an intelligible fashion and written in standard English?

Reviewer #1: Yes

Reviewer #2: Yes

Reviewer #3: Yes

5. Review Comments to the Author

Reviewer #1: line 39

Author may like to add "good" so it reads "... good quality water..."

line 43

Author may need to add "antimicrobial so it reads "... antimicrobial resistant genes..."

lines 53&54

We have recently reported proximal wells of different households can contain genetically indistinguishable strains such that household water is a vehicle for the clonal expansion of resistant bacteria...Include reference!!

Line 62

25 isolates from how many households? The number of households samples would be good to reflect. It's possible this may have been described in the previous publication but this manuscript is a stand alone so you may need to provide a synopsis of key points in the methods.

Line 63

Do you mean to say "municipal areas"?

Line 190 &191

How many household were sampled in total? This is especially for someone who has not and may not be going back to read your previous publication.

Lines 279-286

Are there studies that have reported similar or different findings in well, borehole and/or water from other open sources? It will be helpful to discuss these findings in reference to such findings. If there are such findings then, you may not need to bother. It is good to talk about the implication of these findings but who else is talkin about them? This gives a fair idea of what other researchers are saying about this problem.

Line 318-319

Just wondering if there are studies that have reported such similarities in water sources within similar distances apart.

Line 339-340

Has there been reports that water obtained from these wells and wells around the study are is consumed without treatment? Have there been studies that show water treatment could reduce this risks? Since this water sources are largely private, would you suggest sensitization on water treatment, if this is an effective control measure?

Reviewer #2: I commend the authors for conducting a well-designed and timely study on the transmissible antimicrobial resistance of Escherichia coli from household drinking water. It is my opinion that the integration of WGS and conjugation experiments adds significant value to the understanding of ARG dissemination in environmental reservoirs in Ibadan. The study addresses an important public health concern and contributes valuable insights into the potential risks posed by multidrug-resistant E. coli in drinking water sources especially in the study area. The methodology is well-detailed, and the findings are clearly presented. I particularly appreciate the discussion’s emphasis on public health implications and the need for water treatment before consumption in those areas. Below, I provide some comments and suggestions that may help further strengthen the manuscript.

ABSTRACT

Page 1 Line 16: Do you mean 25 E.coli ISOLATES? Please write this correctly

INTRODUCTION

Lines 53 - 54: It seems you are referring to a previous study that you conducted here. Please provide the citation and reference for this report or study.

METHODS

Lines 62-65: since the readers may not necesarily have access to your previous study, i suggest you provide more information concerning how this data was collected. Also, this concerning data collected in different locations can also be better presented on a map. Were the sampling sites chosen systematically or based on convenience?

what is the justification for the number of samples collected?

Is there a reason why different number of samples were collected at different locations? What was the reason for selecting these differnt locations? was this based on prior contamination reports, random selection or simply because of accessibility? You see why it is important to provide more context about the data?

Line 80: Can you confirm if multiple replicates were performed to ensure reliability?

Line 88: How were sequencing errors or assembly artifacts handled? Was this discussed in the previous study? It would be interesting to know.

Lines 110 - 118: What was the rationale for using solid media for conjugation instead of liquid mating? Also, it is not clear from this description if you included appropriate controls. How did you rule out spontaneous mutations or contamination? This is not clear in your description here. Was a known conjugative strain used as a positive control? just to understand how the conjugation system was validated

RESULTS

Line 135: 100% is not a majority but its the total. Its better to say "All were resistant to erythromycin"

Line 149-150: This explanation should be left for the discussion section

DISCUSSION

Lines 285 - 286: What is the public health implication of this finding? This will be a good place to include the implications here.

Lines 296-298: It is important to state the public health implications of this finding. Also how does this finding compare to studies from other regions with similar water quality challenges?

Lines 338 - 340: What are the strengths of your study? It is important to highlight them here in the discussion.

You have quite profound results, what limitations might affect the interpretation of these results? Could other factors (e.g.Small sample size, biofilm formation, water treatment practices e.t.c) influence the findings? It is important to consider external factors affecting AMR persistence in water systems.

Reviewer #3: I have thoroughly reviewed the manuscript and found no errors or areas that require revision. The study is well-designed, the data are well-documented, and the conclusions are well-supported. The manuscript is clear, coherent, and contributes valuable insights to the field. I have no additional comments or suggestions for improvement.

6. PLOS authors have the option to publish the peer review history of their article (what does this mean? ). If published, this will include your full peer review and any attached files.

**Do you want your identity to be public for this peer review?** For information about this choice, including consent withdrawal, please see our Privacy Policy .

Reviewer #1: No

Reviewer #2: **Yes: ** ABDULHAKEEM ABAYOMI OLORUKOOBA

Reviewer #3: No

---

## [Author Response · Author response to Decision Letter 1]

12 Apr 2025

Response to Review for Manuscript number: PONE-D-25-04412

Title: Transmissible antimicrobial resistance in Escherichia coli isolated from household drinking water in Ibadan, Nigeria

Akeem G. Rabiu, Olutayo I. Falodun, Rotimi A. Dada, Ayorinde O. Afolayan, Olabisi C. Akinlabi, Elizabeth T. Akande and Iruka N. Okeke

Response to the editor

Thank you for the opportunity of peer review and the invitation to prepare a revision. We have responded to and addressed all the reviewers’ comments. The editor was particularly interested in seeing that the following concerns were addressed

1. The editor wants our manuscript to meet PLOS ONE's style requirements. We respond that we have carefully perused the PLOS ONE’s style requirements and complied with the rules accordingly.

2. The editor noted our financial disclosure and wanted a declaration of the role of the funder. We amend the financial disclosure as follows: This work was supported by the African Research Leader’s Award MR/L00464X/1 to INO which was jointly funded by the United Kingdom Medical Research Council (MRC) and the United Kingdom Department for International Development (DFID) under the MRC/DFID Concordat agreement and is also part of the EDCTP2 program supported by the European Union. INO is a Calestous Juma Fellow supported by the Bill and Melinda Gates Foundation (Grant no. INV-036234). The funders had no role in study design, data collection and analysis, decision to publish, or preparation of the manuscript. This has been inserted in the cover letter as advised by the editor.

3. We regret that our study funding information was not appropriately inserted in the online Funding Statement. In the meantime, we have deleted the financial disclosure statement from the Acknowledgement section as advised by the editor. We provide a revised the funding information here that ‘This work was supported by an African Research Leader’s Award to INO, which was jointly funded by the United Kingdom Medical Research Council (MRC) and the United Kingdom Department for International Development (DFID) under the MRC/DFID Concordat agreement and is also part of the EDCTP2 program supported by the European Union. INO is a Calestous Juma Fellow supported by the Bill and Melinda Gates Foundation (Grant no. INV-036234). The funders had no role in study design, data collection and analysis, decision to publish, or preparation of the manuscript.’ This has also been included in the cover letter. The editor is here appreciated for the promise to help reflect the amended funding information in the online Funding Statement.

4. The editor wants the corresponding author to include and validate his ORCiD. We respond that all the authors, including the corresponding have their ORCiD appropriately linked on the manuscript.

5. The editor wants us to review the references and ensure that it is complete and correct and that any retracted papers listed as references are removed. We write that the references are complete and correct, and none of the references has been retracted.

Response to the additional editorial comments

6. The editor wants us to state the key limitation of the study and merge the last paragraph of the discussion section with the conclusion while ensuring that the conclusion appears first. In the revised manuscript, we stated a key study limitation in line 357 that: The small number of isolates examined limited wider generalisation of the import of the study. We have also merged the future directions or recommendations with conclusion but we ensured the initial concluding remarks first appeared as advised by the editor (lines 350-362).

7. The editor observed that 50% of our study’s citations were older than five years, whereas no more than 20% of references are permitted from older sources unless they are foundational studies. We answer that the references have been updated by replacing 11 old references with more recent literature within the past five years and that the older references were limited to key foundational studies.

8. The editor wants us to use the PACE platform to improve the image quality of our figures. We respond that the figures have been improved and converted to .tif using PACE.

Response to reviewer #1

1. We appreciate reviewer #1 for the time spent reviewing the manuscript. Reviewer #1 wants us to add ‘good’ in line 40 to ‘quality water’ so it reads ‘good quality water’. This has been done.

2. The reviewer wants us to add ‘antimicrobial’ to ‘resistance genes’ to read ‘antimicrobial resistance genes.’ This has been done in line 44

3. The reviewer wants us to include a reference at the end of the statement. We have inserted the reference in lines 55-56.

4. The reviewer wants us to state the number of isolates including additional study context. We answer that 25 E. coli isolates were retrieved from 13 households, and this has been inserted in line 76. We provided additional study context under study area subsection in lines 63-72 and further details in lines 74-76.

5. The correction noted by the reviewer has been effected as the phrase now reads ‘municipal areas.’

6. We appreciate the comment of the reviewer concerning additional information on sampling framework. We have supplied more information on sampling context in lines 63-72 and lines 74-76 under the methodology section.

7. The reviewer is appreciated for his/her comments on the implications of our study and for the wishes to amplify this further in the discussion section. We have provided specific resistance genes found in common with other studies in the revised manuscript. We further clarified in the text that these antimicrobial resistance genes are capable of thwarting treatment of infections caused by these strains, especially when first-line antimicrobials are administered, in lines 299-301.

8. The reviewer wondered if there are studies that have reported such similarities in water sources within similar distances apart. We responded that several studies have reported fecal contamination of drinking water in Nigeria1,2 but the current study actually took a step further to have performed the finest resolution of E. coli using whole genome sequencing.

9. The reviewer wants to know if previous reports have shown that water obtained from the wells and wells around the study area are being consumed without treatment and whether treating the water could reduce the risks involved. The reviewer also sought to know whether sensitization on water treatment could effectively reduce the risk associated with the source water contamination since these water sources are largely private. We respond that treatment of drinking water at household level is poorly practised in Nigeria3. However, we agree with the reviewer that household’s sensitisation on treating drinking water, using low-cost point-of-use option4, can be an effective control measure. We have therefore revised this manuscript to reflect this understanding as follows: In the interim, sensitising the householders on the use of ceramic and biosand water filters [35] is recommended to forestall the spread of microbes carrying mobile and transmissible resistance traits.

Response to reviewer #2

1. We are delighted that the reviewer found our narrative interesting and compressive. We thank the reviewer for this supportive comment and appreciate him for his kind and generous comments, and for the time spent in reviewing the article. We provide below point by point responses to his observations/suggestions. Indeed, we agree that attending to the comments of the reviewer have helped to improve the quality of the manuscript’s narrative.

2. The reviewer wants us to insert the word ‘isolates’ immediately after E. coli. This has been done (Line 17).

3. The reviewer wants us to provide the citation and reference for our previous study. We respond that the previous work has been inserted at the end of the sentence in the revised manuscript5 (Lines 55-56).

4. The reviewer wants us to provide additional context on the study area, sample size and collection, and to use map to show where the data were obtained. We appreciate the comments of the reviewer and have inserted study area sub-section under materials and methods to give clear context for the study justification and the sampling framework (lines 63-72, 74-76).

5. The reviewer wants to know if multiple replicates were performed to ensure reliability. We confirm that the samples were analysed in triplicates and that each water sample was collected three times - in a wet season and two dry seasons. We have made this clear in several lines 63-72, 74-76 and 89.

6. The reviewer wants to know how sequencing errors and read artifacts were handled. We respond, in line 104 that: After genome sequencing, sequence reads were demultiplexed and adapters removed and in lines 106-107 that: with contigs <200 bp and coverage <10-fold excluded from the analyses.

7. The reviewer wants to know the rationale for using solid media for conjugation instead of liquid mating and that clarity should be provided on how the controls were handled and the conjugation experiments validated, including how spontaneous mutations or contamination were ruled out. We used solid plate method because it provides high cell density and proximity between donor and recipient cells enabling mating pair formation without the need for mating pair stabilisation which is important for liquid mating method6. As for the control experiment, we indicated in Table 2 of the manuscript that E. coli ATCC 25922 was the control strain. In the revised text, we restated that: Transconjugant colonies consistent with the parent organisms were taken further (lines 136-137) and that: Escherichia coli ATCC 25922 was used as the control strain (line 139).

8. ‘All were resistant to erythromycin’ has been inserted. (line 147).

9. The reviewer prefers that the narration ‘No isolate carried aac-6-lb-cr, which confers ciprofloxacin and aminoglycoside resistance (Fig. 1). Although no carbapenemase genes were detected, nor was resistance to carbapenems, one multidrug-resistant strain carrying mobile colistin resistance gene mcr-1 was found’ should be part of discussion. We appreciate the reviewer’s comments and assure that we have provided a context in the discussion section to reflect the import of the findings but consider that the two statements be permitted as a pretext for a broader view outlined in the discussion. (lines 160-162)

10. The reviewer wants us to state the public health implication of our study. In the revised manuscript, we respond that: It is worrying that these antimicrobial resistance genes are capable of thwarting treatment of infections caused by these strains, especially when first-line antimicrobials are administered. (lines 299-301)

11. The reviewer remarked the importance of stating the public health implications of PMQR and how it compare to studies from other regions with similar water quality challenges. Although we both did not observe quinolones resistance phenotype and QRDR mutations in the strains carrying qnrS1 genes, we, however, explained that the transfer of the PMQR genes can cause clinically significant quinolone resistance in a recipient strain that has them [12]. We have inserted a reference to show how this scenario pans out in the setting (line 311).

12. The reviewer wants us to state the strengths of our study. We thank the reviewer for this comment and answer that our ‘study unveils that without preventing fecal contamination of drinking water, the spread of fecally derived bacteria can support the horizontal spread of ARGs in household water by clonal expansion of resistant strains and transmission of mobile genetic elements’ (lines 354-357).

13. The thank the reviewer for acknowledging that our results are profound. We respond to queries on study limitations as follows: A few isolates examined in this study limited a wider generalisation of the import of the study. Thus, further studies are suggested to determine whether household drinking water can lead to the human acquisition of environmentally disseminated ARGs. This would help to evaluate the extent of the threat to public health posed by consuming drinking water contaminated with bacteria carrying transmissible genes (lines 357-362). We agree with the reviewer that water treatment practices can improve the microbiological quality of the water sources. We therefore suggested, in the revised manuscript, that: sensitising the householders on the use of ceramic and biosand water filters [35] is recommended to forestall the spread of microbes carrying mobile and transmissible resistance traits. (lines 361-362)

Reviewer #3

We thank the reviewer for the time spent in reading the manuscript and the generous kind words and comments on the quality of our work.

References cited in this response to the reviewers

1. Odetoyin B, Ogundipe O, Onanuga A. Prevalence, diversity of diarrhoeagenic Escherichia coli and associated risk factors in well water in Ile-Ife, Southwestern Nigeria. OH Outlook. 2022; 4(1), 3. https://doi.org/10.1186/s42522-021-00057-4

2. Adamu I, Andrade FCD, Singleton CR. Availability of drinking water source and the prevalence of diarrhea among Nigerian households. Am. J. Trop. Med. Hyg. 2022; 107(4), 893–897. https://doi.org/10.4269/ajtmh.21-0901

3. Nwinyi, O.C.., Uyi, O., Awosanya, E.J.., Oyeyemi, I.T.., Ugbenyen, A.M.., Muhammad, A., Alabi, O.A.., Ekwunife, O.I.., Adetunji, C.O.. & Omoruyi, I.M.. Review of Drinking Water Quality in Nigeria: Towards Attaining the Sustainable Development Goal Six. Annals of Science and Technology, 2020, Sciendo, vol. 5 no. 2, pp. 58-77. https://doi.org/10.2478/ast-2020-0014

4. Erhuanga E, Banda MM, Kiakubu D, Kashim IB, Ogunjobi B, Jurji Z, et al. Potential of ceramic and biosand water filters as low-cost point-of-use water treatment options for household use in Nigeria. J. Water Sanit. Hyg. Dev. 2021; 11 (1): 126–140. doi: https://doi.org/10.2166/washdev.2020.096

5. Rabiu AG, Falodun OI, Fagade OE, Dada RA, Okeke IN. Potentially pathogenic Escherichia coli from household water in peri-urban Ibadan, Nigeria. J Water Health. 2022;20(7):1137–49. http://dx.doi.org/10.2166/wh.2022.117

6. Allard, N., Collette, A., Paquette, J. et al. Systematic investigation of recipient cell genetic requirements reveals important surface receptors for conjugative transfer of IncI2 plasmids. Commun Biol 6, 1172 (2023). https://doi.org/10.1038/s42003-023-05534-2

---

## [Decision Letter · Decision Letter 1]

29 Apr 2025

Transmissible antimicrobial resistance in Escherichia coli isolated from household drinking water in Ibadan, Nigeria

PONE-D-25-04412R1

Dear Dr. Falodun,

We’re pleased to inform you that your manuscript has been judged scientifically suitable for publication and will be formally accepted for publication once it meets all outstanding technical requirements.

Kind regards,

Mabel Kamweli Aworh, DVM, MPH, PhD. FCVSN

Academic Editor

PLOS ONE

Additional Editor Comments (optional):

Reviewers' comments:

Reviewer's Responses to Questions

**Comments to the Author**

1. If the authors have adequately addressed your comments raised in a previous round of review and you feel that this manuscript is now acceptable for publication, you may indicate that here to bypass the “Comments to the Author” section, enter your conflict of interest statement in the “Confidential to Editor” section, and submit your "Accept" recommendation.

Reviewer #1: All comments have been addressed

Reviewer #2: All comments have been addressed

2. Is the manuscript technically sound, and do the data support the conclusions?

Reviewer #1: Yes

Reviewer #2: Yes

3. Has the statistical analysis been performed appropriately and rigorously? 

Reviewer #1: N/A

Reviewer #2: Yes

4. Have the authors made all data underlying the findings in their manuscript fully available?

Reviewer #1: Yes

Reviewer #2: Yes

5. Is the manuscript presented in an intelligible fashion and written in standard English?

Reviewer #1: Yes

Reviewer #2: Yes

6. Review Comments to the Author

Reviewer #1: The author has addressed all the comments identified during my review. The manuscript presents a good addition to the body of knowledge of water contamination and water safety.

Therefore, I recommend it for publication.

Reviewer #2: Authors have satisfactorily responded to all my comments and questions and can now proceed with the publication process.

7. PLOS authors have the option to publish the peer review history of their article (what does this mean? ). If published, this will include your full peer review and any attached files.

**Do you want your identity to be public for this peer review?** For information about this choice, including consent withdrawal, please see our Privacy Policy .

Reviewer #1: **Yes: ** Igbaver Ieren

Reviewer #2: **Yes: ** ABDULHAKEEM Abayomi OLORUKOOBA

---

## [Editor Report · Acceptance letter]

PONE-D-25-04412R1

PLOS ONE

Dear Dr. Falodun,

I'm pleased to inform you that your manuscript has been deemed suitable for publication in PLOS ONE. Congratulations! Your manuscript is now being handed over to our production team.

Kind regards,

on behalf of

Dr. Mabel Kamweli Aworh

Academic Editor

PLOS ONE